# Microstructure of Deposits Sprayed by a High Power Torch with Flash Boiling Atomization of High-Concentration Suspensions

**DOI:** 10.3390/ma17071493

**Published:** 2024-03-25

**Authors:** Saeid Amrollahy Biouki, Fadhel Ben Ettouil, Andre C. Liberati, Ali Dolatabadi, Christian Moreau

**Affiliations:** 1Department of Mechanical, Industrial and Aerospace Engineering, Concordia University, Montreal, QC H3G 1M8, Canada; 2Department of Mechanical and Industrial Engineering, University of Toronto, Toronto, ON M5S 3G8, Canada

**Keywords:** flash boiling atomization, suspension plasma spray, high concentration suspension, high plasma torch power

## Abstract

The main objective of this study was to use flash boiling atomization as a new method to inject suspensions with high solid content into the high-power plasma flow. The water-based suspension was prepared with submicron titanium oxide particles with an average size of 500 nm. The investigated solid concentrations were 20, 40, 55 and 70 wt%. Two plasma torches operated at 33, 70 and 110 kW were used to investigate the effect of increasing power on the deposited microstructure and deposition efficiency. At low torch power, the deposition efficiency decreased with increasing solid concentration, and deposits with a high number of unmelted particles were obtained with 70 wt% suspensions. At high torch power, the deposition efficiency increased with increasing solid concentration, and dense deposits were obtained with 70 wt% suspensions. XRD analysis was performed on all deposits to determine the distribution of rutile and anatase phases. The percentage of the anatase phase varied from 35.7% to 66.9%, depending on the power input and solid concentration.

## 1. Introduction

Suspension plasma spraying (SPS) is a form of plasma spraying that allows for the injection of nano-sized or submicron-sized powders into a plasma jet. These small particles are dispersed in a solvent to give them enough momentum to enter the plasma jet. The solvent is rapidly vaporized when exposed to the plasma stream, and the small particles are melted, accelerated and directed to impact the substrate and form a coating. A unique coating microstructure can be created with this method, and both the mechanical and thermal behavior of coatings can be improved by using SPS compared to conventional air plasma spray (APS) coatings [1,2]. However, the feed rate of the powders and deposition efficiency in SPS are typically two to three times lower than in APS [3]. The use of a suspension with higher particle concentration is one way to improve the deposition of the feedstock. However, clogging can be an obstacle to injecting higher concentration suspensions (above 40 wt%) because increasing the particle concentration increases the viscosity of the suspension, which leads to a decrease in suspension flowability. Nevertheless, flash boiling atomization has been found to be an effective method for transferring and injecting high-viscosity fluids in various industries [4,5].

Flash boiling is the thermodynamic phenomenon that occurs when a pressurized superheated liquid (i.e., a liquid heated to temperatures above its boiling point without boiling) is exposed to an environment where the ambient pressure is lower than the saturation pressure of the liquid. Flash boiling atomization has various applications ranging from fuel injection in combustion chambers [6] to distillation [7], spray drying [8] or the pharmaceutical industry [9]. Since high temperatures help to reduce the viscosity of liquids or solutions, this method is also used to spray highly viscous liquids. Karami [10] used flash boiling atomization to spray black liquor (pulpwood to paper pulp byproduct) and studied the spray parameters, such as the spray pattern, at different injection temperatures. The viscosity of the black liquor at 70 wt% solids and 127 °C is 88 times higher than the viscosity of water at room temperature [10]. Gunther et al. [11] used flash boiling to spray a polyvinylpyrrolidone solution and investigated the spray characteristics at different temperatures, nozzle geometries and fluid properties. This type of polymer has a viscosity 25 times greater than water at a solid concentration of 20 wt%.

Suspension injection in SPS can be either axial or radial. In axial injection, the suspension is fragmented inside the torch by interaction with the atomizing gas, and the fragmented droplets are then exposed to the plasma jet. In radial injection, the suspension is injected into the plasma stream from outside the torch. Radial injection allows for more control over the injection parameters, such as the injection angle or adjusting the distance between the nozzle tip and the plasma jet. Mechanical injection and spray atomization are two different radial injection methods currently used in SPS.

In mechanical injection, a single continuous jet is injected into the plasma jet using a pressurized gas, and different jet breakup regimes occur in the plasma jet based on the Weber number of the gas (defined as the ratio between inertial forces and surface tension forces). In spray atomization, the suspension is injected as droplets generated by an atomizer. Toma et al. [12] used two different injection methods, mechanical injection and spray atomization (using an inert gas for atomization), to inject fine TiO_2_ particles for the preparation of photocatalytic titania coatings. They showed that the microstructure of the coatings did not change for the different suspension injection methods. However, they found that the deposit thickness per pass decreased from 4–6 µm for mechanical injection to 1.5–2 µm for spray atomization. Three disadvantages of spray atomization over mechanical injection have been reported in the literature: First, gas atomization spraying creates disturbances in the plasma jet. Second, it would be more difficult to achieve precise injection of the feedstock at the chosen location of the plasma jet with this spraying method. Third, there is a dispersion of droplet trajectories, sizes and velocities when working with spray atomization [13].

In mechanical injection, the injection pressure is the most important injection parameter that can affect the deposition efficiency rate, coating properties and microstructure. Meillot et al. [14] studied the feedstock penetration at different injection pressures for two different injection modes (up-to-down and down-to-up). They found that the feedstock required an optimal high pressure to reach the plasma center, while a very high pressure would lead to decreasing deposition efficiency. Fauchais et al. [15] studied the effect of injection pressure and concluded that by increasing the pressure, the suspension can penetrate deeper to reach the hot region within the plasma jet. In spray atomization, in addition to injection pressure, other parameters such as nozzle design [16] and ALR (gas-to-liquid mass ratio) [17] can affect the size and velocity of the droplets and the properties of the coating. Marchand et al. [16] designed two different internal and external two-fluid nozzles to spray yttria-stabilized zirconia (YSZ) into the plasma jet. They achieved different sprays with different droplet sizes and velocity distributions by changing the design of the atomizing nozzle. Rampon et al. [17] showed different spray patterns of yttria partially stabilized zirconia (Y-PSZ) suspension for three ALR values (0.15, 0.3 and 0.6). It was observed that better atomization occurred at higher ALR. However, the injection of atomizing gas into the plasma jet leads to a decrease in plasma temperature and consequently reduces the heat transfer to the suspension droplets. In flash boiling atomization, no gas is required for atomization and vapor bubbles are responsible for breaking the suspension jet into small droplets, so this method can overcome the adverse effects of using atomization gas on the plasma flow, such as plasma cooling and plasma disturbance. 

In flash boiling atomization, not only is no atomization gas added to the plasma jet, which contributes to plasma cooling, but the disintegrated droplets at the nozzle exit can also reach temperatures close to the water boiling point. This indicates that approximately 20% of the energy required for water evaporation with the plasma jet can be saved, as 80% of the required energy for water evaporation is associated with latent heat. In our previous work, we compared two different coatings fabricated using flash boiling atomization and a single jet method. Our results demonstrate that flash boiling atomization can achieve slightly higher deposition with wider deposits compared to the conventional single-jet method [18].

This study focuses on the preparation and injection of a high-concentration submicron-sized titania suspension into a high-power plasma jet to develop SPS deposits. The spray parameters were selected to ensure stable plasma conditions and a controlled deposition process. To investigate the influence of increasing suspension concentration on coating microstructure, thickness, and deposition weight, four different solid concentrations were used: 20, 40, 55 and 70 wt%. The effect of torch power on the microstructure of the deposits, as well as on the deposition efficiency of the process, was investigated by using three different power levels: 33, 70 and 110 kW. The effect of torch power and solid concentration on the phase composition of the deposits was determined via XRD analysis.

## 2. Materials and Methods

### 2.1. Experimental Setup and SPS Conditions

The experimental setup is shown in Figure 1. Two tanks were used: one for water (1) and the other for suspension (2). To avoid using more suspension, the experiment started by initially introducing water to the line to generate a consistent flash boiling spray. Then, the plasma torch was activated, and the three-way valve (4) was opened to introduce the suspension into the line. Water was also used to clean the line after each injection. The suspension was continuously agitated with a magnetic stirrer (3) during the experiment to prevent sedimentation. The air injection pressure (5) was 0.69 MPa. The flow rate of the suspension was measured using a Coriolis flow meter (6) placed before the heater. This type of flow meter (Proline Promass 83, Endress+Hauser, Basel, Switzerland) can monitor the suspension feed rate and the density of the suspension. The suspension was then superheated to 140 °C in the duct heater (7) but remained in the liquid phase because it was pressurized. The power of the heater can be regulated from 0 to 1.2 kW by connecting it to a variac transformer (8), and the desired temperature can be obtained by changing the power. The temperature was continuously monitored using a thermocouple (9) inserted at the outlet of the heater. The heater was connected to the torch nozzle holder (11) by a flexible plastic tube (10). This tube allows for more flexibility in the movement of the robotic arm. A cylindrical stainless-steel nozzle (12) with a diameter of 0.15 mm and a length of 1.25 mm was attached to the robot arm with a nozzle holder.

In this experimental setup, the movement of the robotic arm was limited and did not allow for the entire substrate to be coated. The torch was attached to the robotic arm and held stationary in front of a rotating sample holder (13). The stand-off distance (distance between the torch exit and the substrate) was 50 mm. Stainless-steel substrates with dimensions of 25 mm × 50 mm were grit blasted with 80-grit alumina particles (i.e., particle size of 160 µm) and were then mounted on the rotating sample holder. Two different rotational speeds of 60 and 180 rpm (with linear velocities of 1 and 3 m/s, respectively) were used for the sample holder. Different spray times were chosen based on the suspension concentration because it was observed that delamination can occur at spray times longer than one minute.

The 3 MB torch (Oerlikon-Metco, Pfaffikon, Switzerland) and the Mettech Axial III torch (Northwest Mettech Corp., North Vancouver, BC, Canada) were utilized to produce deposits using low and high plasma powers, respectively. In both cases, the suspension was injected radially into the plasma jet downstream of the torch nozzle exit. The spray conditions for the 3 MB and Axial III torches are shown in Table 1 and Table 2, respectively. The Axial III torch is designed for axial injection; therefore, some modifications were required to allow for radial injection. The plasma power was 33 kW for the 3 MB torch, and two plasma powers of 70 kW and 110 kW were used for the Axial III torch. The vertical distance between the nozzle tip and the center of the plasma jet was 8 mm for the 3 MB torch and 15 mm for the Axial III torch.

The substrate temperature was not measured directly, but based on several previous experiments, it was known that sample temperatures could not exceed 300 °C [19] for the 3 MB torch and 450 °C for the Axial III torch when using a rotating sample and not preheating the substrates [20]. The repeatability of experiments was confirmed by repeating each experiment two or three times, depending on the experimental conditions.

### 2.2. Suspension and Feed Material

The suspension was prepared by dispersing submicron-sized titanium dioxide particles (TKB Trading, Oakland, CA, USA) with an average size of 500 nm in deionized water. A SEM micrograph of the SPS feedstock powder is presented in Figure 2. The particle concentrations were 20, 40, 55 and 70 wt%. To produce uniform suspensions, especially at high concentrations (more than 70 wt%), the particles were added slowly to the water and mixed simultaneously using a magnetic stirrer and an ultrasonic liquid mixer (QSonica, Newtown, CT, USA). The suspension injection conditions used in this study are detailed in Table 1 and Table 2. 

In this research, nano titanium dioxide powder was chosen for its long-term sustainability, accessibility and cost-effectiveness. Titanium dioxide (TiO_2_) coatings are mostly recognized for their photocatalytic properties, and they are widely used in photocatalytic applications such as air and water purification, wastewater treatment and self-cleaning surfaces across various industries [19].

The viscosity of the suspension increases significantly with increasing solid concentration [21], and the fluidity of the suspension also decreases in this condition. For this specific powder, Garmeh [22] conducted experiments to generate stress–strain rate curves for identical TiO_2_ suspensions with varying concentrations of 30, 50 and 70 wt%. The findings showed a notable rise in suspension viscosity when the concentration increased from 50% to 70%. In a highly viscous suspension, due to the presence of more solid particles, the interaction between the particles would be increased, and agglomeration occurs. An efficient method for decreasing the viscosity of the suspension involves using an appropriate dispersing agent in relation to the mixture [23]. An alternative approach chosen for this experiment entails increasing the temperature of the suspension until it reaches a superheated condition. The viscosity of a suspension is inversely proportional to the temperature, and the viscosity decreases significantly at high temperatures; furthermore, the flowability would be improved in this condition [11]. It has been observed that suspensions with high TiO_2_ concentration can be injected with this method for a longer injection time without clogging.

### 2.3. Visualization and Characterization Measurements

The flash boiling spray was captured using the shadowgraph technique with a high-speed camera. The vertical distance between the suspension injector tip and the plasma jet was chosen based on these shadowgraph results. This vertical distance is important to ensure that most disintegrated droplets can penetrate the plasma jet. The microstructure of the generated deposits was observed using scanning electron microscopy (SEM) (Hitachi S3400, Tokyo, Japan) at different magnifications. In addition, the phase identification (rutile and anatase distribution) of the deposits was measured via X-ray diffraction (XRD) using a X′ Pert Pro; PANalytical (Malvern, UK), (Philips, Amsterdam, The Netherlands). Cu-Kα radiation was used, and the 2θ angle ranged from 10 to 90°with a step size of 0.02°.

### 2.4. Flash Boiling Atomization Condition

Flash boiling atomization is a thermodynamic process that can produce a spray of droplets. When a pressurized suspension (at a pressure of 0.69 MPa) passes through the duct heater, its temperature rises from room temperature to over 140 °C, which has become a superheated suspension. In this state, the suspension can be kept in the liquid phase due to its high pressure, but the maximum temperature must be kept below the boiling temperature of the injection pressure (the boiling temperature at 0.69 MPa is 169 °C) because the suspension may evaporate inside the heater. When the superheated suspension passes through a nozzle, it experiences a rapid pressure drop from injection pressure to atmospheric pressure (0.1 MPa). At the time of pressure reduction, when the pressure reaches the saturation pressure of the injection temperature (a saturation pressure at 140 °C is 0.27 MPa), the suspension cannot be maintained as a liquid phase, and the phase change begins.

The onset of phase change occurs at nucleation sites, and the vapor bubbles begin to emerge from these sites. In pure water without impurities and dissolved gas, the wall of the nozzle is the main nucleation site. In suspension, in addition to the wall roughness, the surface of TiO_2_ particles can promote the formation of steam bubbles. These vapor bubbles start growing rapidly, eventually reaching a point of expansion where they finally burst, breaking the suspension jet into smaller droplets. The flash-boiling atomized spray jet of titanium dioxide suspension is shown in Figure 3.

In the flash-boiling spray jet, larger droplets remain in the center, while smaller droplets are visible at the edge of the spray jet. The larger droplets at the center of the spray have enough momentum and ate in the right direction to reach the hot region of the plasma jet, which can create a well-melted zone in the deposited microstructure. However, small droplets at the edge of the spray may be driven away by the plasma flow or may only penetrate to the outer zones of the lower-temperature plasma flow, creating an unmelted zone in the deposits.

## 3. Results and Discussion

### 3.1. Microstructure of TiO_2_ Deposits Fabricated with High Torch Power

The cross-sectional microstructure of the TiO_2_ deposit obtained through flash-boiling spray with a high-power torch (110 kW) and low solids concentration is illustrated in Figure 4. A bimodal microstructure can be observed for the deposit: fully melted particles (light region) and unmelted particles (gray region). In addition, some dark areas correspond to the porosity observed in the deposit, which is negligible compared to other sections. 

In our earlier investigation [18], we presented the deposit formed under the conditions of C1, characterized by lower power (3 MB torch at 33 kW, solid concentration of 20 wt%), which exhibited a high number of unmelted areas mixed with well-melted particles. Notably, there was an observable gradient of unmelted particles within the deposit, with a high concentration near the deposit–substrate interface. This may be related to a rapid cooling rate between the stainless-steel substrate and the high-temperature particles. In our experimental setup (Figure 1), it is not possible to preheat the substrates because it takes several minutes to obtain the desired flash boiling spray jet, and the torch cannot be ignited in the meantime.

In contrast, the deposit formed under the conditions of C6 (Figure 4), produced at higher power (Axial III torch, solid concentration of 20 wt%), exhibited a more uniform deposit with a higher proportion of well-melted particles. By increasing the power of the torch from 33 kW to 110 kW, more energy is available for liquid evaporation and particle melting. Under these conditions, unmelted particles are not only concentrated near the substrate, but mixtures of well-melted and unmelted particles are also distributed throughout the entire deposit.

Figure 5 shows a higher magnification image of the TiO_2_ deposit under the conditions of C6. The deposit comprises distinct zones: some are formed from the solidified splats resulting from the impact of fully melted particles, while others consist of unmelted particles coming from the suspension itself, with diameters ranging from 100 nm to 300 nm. A comparison between the unmelted particles within the deposits and the starting powder reveals that certain initial particles remain unmelted and are unable to reach the hot region in the plasma jet. In radial injection, unlike the case of axial injection, the heat transfer between the plasma jet and the TiO_2_ particles is not sufficient to provide the particle melting energy. As a result, the deposit produced with radial suspension injection and lower torch power is mainly built up by unmelted particles impacting the substrate. The non-melted zones have a weaker bond strength, which can be easily removed from the deposit with less abrasion [24]. In addition, the concentration of unmelted particles at the deposit-substrate interface for low-power torch can lead to weak deposit adhesion, and tolerance for thick deposits would not be supported. This delamination is the primary failure mode observed for deposits produced at low torch power and long spray times (greater than one minute at 60 rpm).

### 3.2. Deposits Fabricated with High Solid Content Suspensions

Figure 6 shows deposit microstructures produced by using suspensions containing 70 wt% of TiO_2_ particles and spraying at two different torch powers (33 kW and 110 kW): these are deposits C4 (Figure 6a) and C7 (Figure 6b). At low torch power, a significant number of unmelted particles can be observed by increasing the solids concentration from low suspension concentrations to 70 wt% (Figure 6a). In addition to the lower power available for particle melting, two other reasons could explain this high number of unmelted particles. First, in suspensions with high solid concentrations, there is high particle interaction, which tends to facilitate particle agglomeration. These agglomerates, as well as larger particles, remain partially or completely unmelted in the plasma jet, and a higher plasma torch is required to melt these agglomerates; therefore, many unmelted particles are obtained under low-power conditions [25]. This is illustrated in Figure 7, which shows a magnified cross-section of a 70 wt% suspension deposit, showing a large number of agglomerates in the deposited microstructure. Suspension agglomeration may happen under each of the suspension conditions; however, when the suspension concentration is high, there is a greater chance for particles to come into close proximity, leading to increased agglomeration. In addition, the higher concentration means more solid particles are present in a specific volume of suspension, increasing the chance of collision to make an agglomeration.

Second, flash-boiling atomization produces less spray atomization (primary atomization) with a larger droplet size distribution in high-viscosity suspensions because the high viscosity hinders bubble growth and bubble bursting inside the suspension [11]. In addition, the surface tension of the suspension would increase with the solid concentration, leading to a decrease in the Weber number (Equation (1)) and, consequently, less secondary breakup of large droplets occurs inside the plasma jet. The fragmentation of the suspension inside the plasma jet is strongly dependent on the Weber number, and complete breakup usually occurs at high Weber numbers [26]. Less primary and secondary fragmentation results in the presence of larger suspension droplets inside the plasma jet and, ultimately, a higher number of unmelted particles in the microstructure of deposits.
(1)$$\mathrm{We}=\frac{\rho_{g}\times u_{r}^{2}\times d_{l}}{\sigma_{l}}$$
where $\rho_{g}$ is the mass gas density (kg/m^3^), $u_{r}$ is the relative velocity between the gas and liquid (m/s), $d_{l}$ is the drop or liquid jet diameter (m), and $\sigma_{l}$ is the liquid surface tension (N/m).

However, a dense deposit microstructure can be achieved at high solid concentrations and high torch power (110 kW) (Figure 6b). The heat transfer increases by increasing the torch power, and more energy is available for particle and agglomerate melting. In addition, the gas velocity and momentum are increased with higher torch power [19], and the Weber number is proportional to the square of the velocity, resulting in increased breakup within the plasma stream. In contrast to the low torch power, high torch power produces a denser deposit by increasing the solid concentration from low suspension concentrations to 70 wt%. Vicent et al. [27] calculated the power required to plasma spray water suspensions of alumina/titania by SPS at different solid contents of the suspension feedstock and showed that this power is reduced by increasing the solid concentration. Therefore, by increasing the concentration, a smaller amount of water is injected along the plasma flow, and the energy consumption related to water evaporation during the plasma jet would be reduced.

Regardless of torch power, no cracks are observed in the deposits produced with low solid concentrations. However, many horizontal and vertical cracks can be found at the boundaries between unmelted and well-melted particles in the high-concentration, low-torch-power deposits (Figure 6a) because these sections are the weakest part of the deposits. In addition, vertical cracks appeared in the high solid concentration and high torch power deposits (Figure 6b). These vertical cracks can enhance the strain tolerance of the coatings. When the coating is subjected to mechanical stresses or strains, these cracks help distribute the stress more evenly by deflecting and absorbing energy. Additionally, they provide pathways for the dissipation of mechanical energy.

In these experiments, poor-quality deposits with many cracks were obtained when spraying under C5 conditions, i.e., with solids concentrations above 55 wt%, torch power above 70 kW, and a lower speed of 60 rpm (Figure 8). This could be explained by the high solid deposition per pass at low rotational speeds, and thus, the thermal expansion of the molten particles, as well as the thermal stress caused by the large temperature differences between the deposited feedstock and the substrate, can destroy the deposit structure [28]. At high solid concentrations/high torch power, the rotational speed was tripled to avoid the destruction of the deposits.

### 3.3. Deposition Efficiency

In our experiments, flash-boiling atomization demonstrated the ability to inject high solid concentrations without any clogging issues. In suspensions with a high concentration, a significant portion of the suspension consists of particles, resulting in a substantial number of particles being injected into the plasma jet. Figure 9 illustrates the amount of TiO_2_ powder injected toward the substrate under various plasma spraying conditions. Comparing the injected particles under different spray conditions reveals a significant increase in the number of injected particles with an increasing solids concentration. This relationship is not linear; for instance, the number of injected particles per second for C4 (70 wt%) is approximately five times higher than that for 20 wt% solids content. However, it is important to note that each spray condition exhibits varying deposition efficiency, meaning that only a portion of this injected powder may be coated onto the substrates.

Figure 10 shows the deposition efficiency of deposits produced via flash boiling atomization under different conditions. The deposition efficiency is defined by dividing the change in weight of the substrates before and after spraying by the weight of TiO_2_ powder used during spraying. The deposition efficiency for condition C1 (20 wt% at 33 kW torch power) is 35% and decreases with increasing solid concentration at low torch power (33 kW), reaching 26% for C2 (40 wt%), 25% for C3 (55 wt%), and finally 21% for C4 (70 wt%). Bernard et al. [29] observed that when the plasma enthalpy is low, the deposition efficiency is reduced by injecting more material into the plasma jet. In addition, Carnicer et al. [30] observed similar results, i.e., that increasing the particle concentration resulted in a decrease in deposition efficiency. Therefore, it appears that as a direct result of increasing the solid concentration at low torch power, less energy is available from the plasma jet to melt individual particles, resulting in lower deposition efficiencies.

However, it was also observed that increasing the solid content at high torch power would increase the deposition efficiency (Figure 10). In particular, the deposition efficiency increased from 46% for C6 (20 wt% at 110 kW torch power) to 60% for C7 deposit (70 wt%). This is consistent with the findings of Curry et al. [31], who also showed that increasing the solids content at a high torch power of 105 kW would increase the deposition efficiency. In addition, the deposition efficiency was significantly improved by increasing the torch power at constant solids concentration. For example, the deposition efficiency is 25% for deposit C3 (33 kW) and increases to 61% for deposit C5 (70 kW power) (Figure 10). The reason for these high deposition efficiencies can be related to the higher input power, which not only compensates for the energy lost in water evaporation but also allows for more complete particle melting.

The deposition efficiency of SPS is about 20%, whereas the deposition efficiency of other conventional spray techniques can reach 55–80% [32]. Furthermore, the deposition efficiency for water-based suspensions is half that of ethanol-based suspensions due to the high evaporation heat of water [32]. However, it has been found that high deposition efficiencies (up to 60%) can be achieved for SPS using flash boiling atomization, especially when using a high-concentration suspension, a high-power torch, and a high rotational speed of the sample holder. This highlights the added value of using flash boiling atomization.

### 3.4. Phase Composition

The XRD analysis of the TiO_2_ deposits under various spraying conditions is depicted in Figure 11. This figure illustrates the XRD pattern for all deposits generated at high torch powers (Axial III torch at 70 and 110 kW). All of the deposits were identified as mixtures of rutile and anatase phases. The percentage of each phase for both low- and high-power torches was determined using the relative peak intensity and is listed in Table 3. The percentage of the anatase phase varies from 35.7% to 66.9%, depending on the suspension properties and plasma conditions. The coatings produced under low torch power settings (C1 to C4) exhibit a higher concentration of anatase in comparison to rutile, as indicated in Table 3. This disparity can be attributed to the lower plasma power and, subsequently, lower plasma temperatures.

Under low torch power, more anatase phase would be formed by increasing the solid content, and the anatase distribution was increased from 59.4% for C1 (solid concentration is 20 wt%) to 66.9% for C4 (solid concentration is 70 wt%). Additionally, it seems that when the anatase phase of TiO_2_ nanoparticles was higher, a large number of unmelted particles was observed in the deposit microstructure, which can be observed in the SEM images (Figure 6a). There is a suggestion that the anatase phase is linked to partially and unmelted particles, whereas the rutile phase is ascribed to particles that have undergone complete melting [33,34].

Our measurements revealed that the main peak for the anatase phase when using a low-power torch can be found between 2θ = 25.2° and 2θ = 25.4° depending on the different solid concentrations, which is in agreement with the peak values reported in the literature [35]. This can be explained by the low torch power, for which there is no significant change in the peak intensities related to the anatase and rutile phases. The main peak for the rutile phase occurs at 2θ = 27.8° at a low plasma temperature. Under the conditions of C4, the deposit formed shows an anatase-to-rutile ratio of approximately 2:1, the highest proportion of anatase to rutile of all other conditions. This could prove valuable for photocatalytic utilization because the anatase phase can exhibit greater photocatalytic behavior compared with the rutile phase [36].

In contrast to the low torch power, the rutile phase is predominantly formed, and the irreversible phase transformation of TiO_2_ nanoparticles from anatase to rutile is more prevalent. This can be explained by the high plasma temperature, which is sufficient to cause excessive heating of the particles, which can promote this phase transformation. In contrast to the lower-power torch, where increasing the solid concentration resulted in less rutile and more anatase formation, the higher-power torch resulted in an increase in the rutile phase from 51.9% for deposit C6 (20 wt%) to 63.9% for deposit C7 (70 wt%). A high solid concentration in the suspension promotes the agglomeration of particles injected into the plasma jet. The heat transfer rate between the plasma flow and the agglomerates would be much lower than the heat transfer rate with the nanoparticles; therefore, depending on their size, the agglomerates may not be melted or even partially melted inside the plasma jet. Thus, the higher-power torch is required to convert the agglomerates from the anatase phase to the rutile phase. The rutile content for sample C4 (70 wt% and 33 kW torch power) is 33.1%, and this amount is approximately doubled to 63.9% for sample C7 (70 wt% and 110 kW torch power). 

On the other hand, at a low solid content (20 wt%), no significant increase in the rutile phase was observed by increasing the torch power from 40.6% for sample C1 (33 kW) to 51.9% for sample C6 (110 kW). When using flash-boiling atomization for SPS, the droplets generated from the suspension should reach the core region of the plasma jet (hot zone) to achieve successful deposit deposition. This condition is more easily achieved for suspensions with high solids concentrations (e.g., 70 wt%) because their density and momentum can be up to twice that of suspensions with lower solids content (Table 1).

## 4. Conclusions

Flash boiling atomization was applied as a new method of suspension injection in suspension plasma spraying to inject suspensions with high solid content. In this study, a water-based TiO_2_ suspension with a particle size of 500 nm was used with different solid concentrations (20, 40, 55 and 70 wt%). In this method, a pressurized suspension is first superheated to temperatures above 140 °C via a duct heater, then injected through a 0.15 mm stainless steel nozzle, and the spray jet of disintegrated droplets penetrates into the plasma jet. Unlike other spraying methods, no additional gas is added to the suspension for atomization, and the vapor bubbles created in the capillary area of the nozzle by rapid depressurization are responsible for atomization. A low-power torch at maximum power (33 kW) and a high-power torch at two power levels (70 and 110 kW) were used to investigate the effect of torch power on deposit properties such as deposit microstructure, deposition efficiency and phase composition. The key findings from our experimental investigation are presented below:(1)Remarkably high deposition rates could be achieved with the high-power torch and by increasing the solids concentration from 20 wt% to high concentrations such as 55 and 70 wt%.(2)Deposition efficiency decreases with increasing solid concentrations at low torch power because less energy is available for particle melting under low-power conditions. Conversely, at high torch power, increasing solid concentrations enhances deposition efficiency due to better penetration into the hot region of the plasma jet.(3)Using suspensions containing 70 wt% of feedstock and varying torch power from 33 kW to 110 kW, a variety of microstructures ranging from dense to highly porous have been achieved. Low torch power and high solids concentration can produce large amounts of unmelted particles, but high torch power and high solids concentration can produce a very dense deposit (mostly composed of well-melted particles).(4)XRD analysis showed that at low torch power, the anatase content is higher than the rutile content, and the anatase percentage increases with increasing solid concentrations. However, the anatase phase is less dominant at high torch power and would decrease with increasing solid concentrations.

## Figures and Tables

**Figure 1 materials-17-01493-f001:**
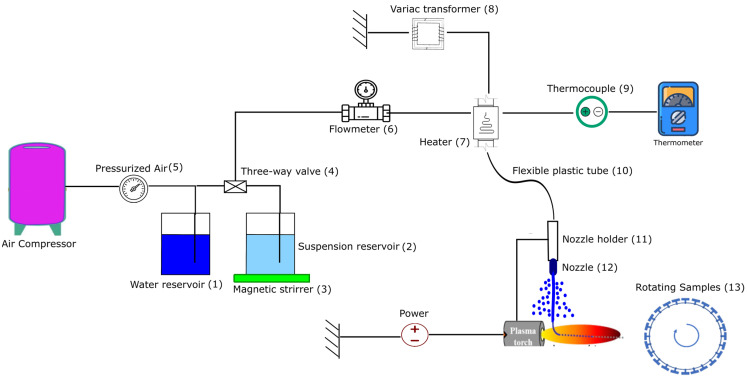
Schematic of the experimental setup [18].

**Figure 2 materials-17-01493-f002:**
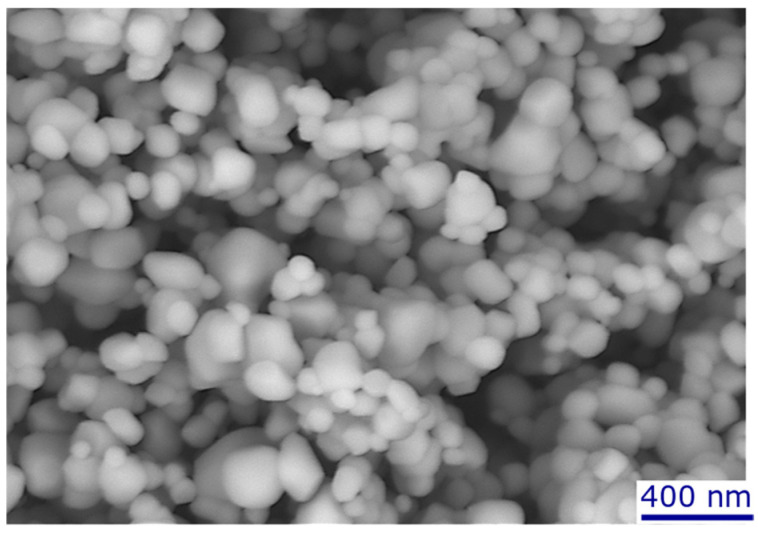
SEM micrograph of SPS feedstock TiO_2_ powder.

**Figure 3 materials-17-01493-f003:**
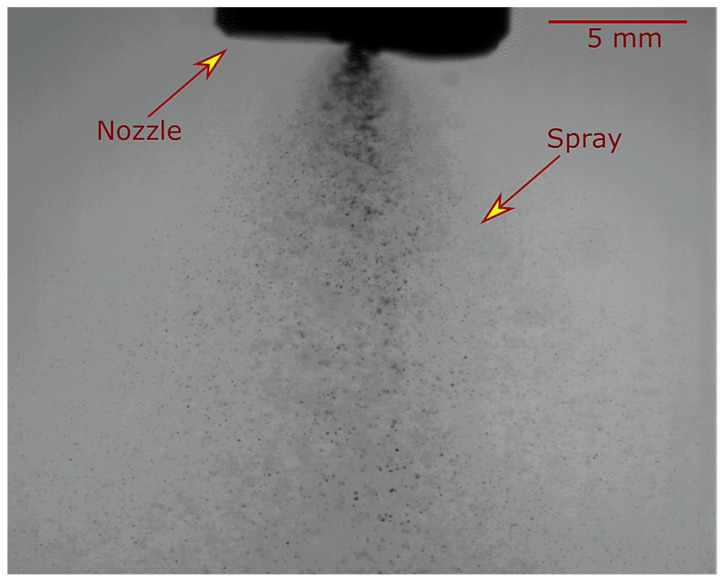
Spray jet morphology of the flash boiling atomization of the TiO_2_ suspension. Suspension concentration: 20 wt% (condition C1).

**Figure 4 materials-17-01493-f004:**
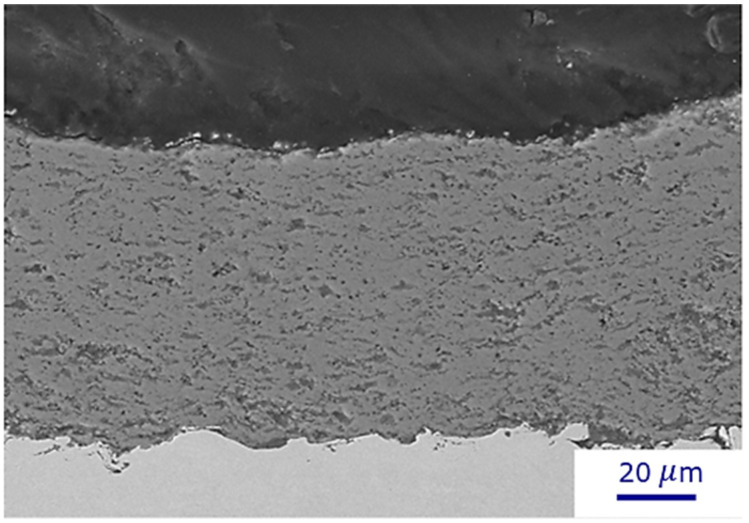
SEM micrographs of TiO_2_ deposits obtained by flash-boiling atomization for a suspension concentration of 20 wt% and plasma power of 110 kW (condition C6).

**Figure 5 materials-17-01493-f005:**
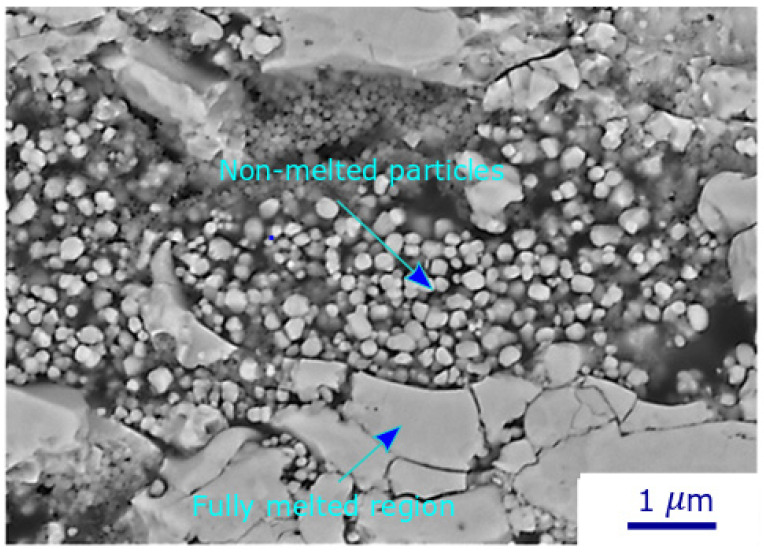
High-magnification SEM micrograph showing non-melted particles and fully melted regions in a TiO_2_ deposit (condition 6).

**Figure 6 materials-17-01493-f006:**
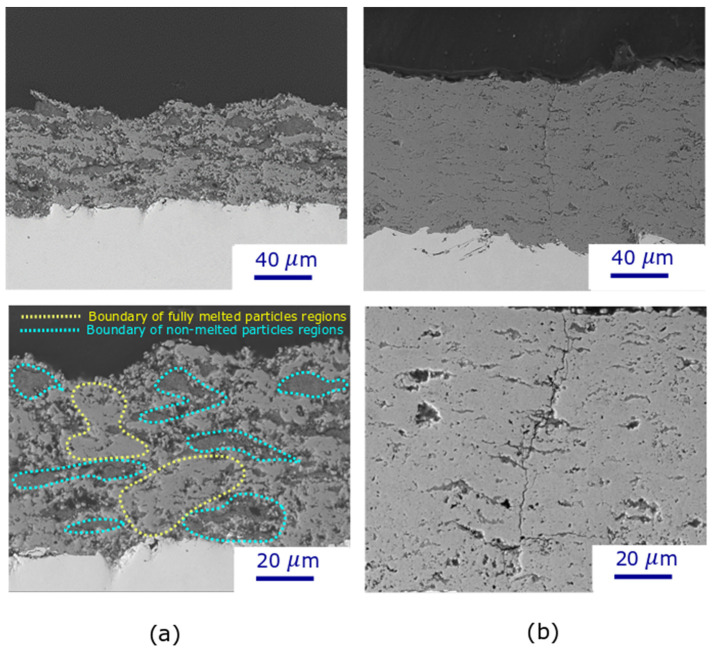
SEM micrographs of TiO_2_ deposits obtained via flash-boiling atomization at different torch powers and solids content concentrations (**a**) C4 [18] (**b**) C7.

**Figure 7 materials-17-01493-f007:**
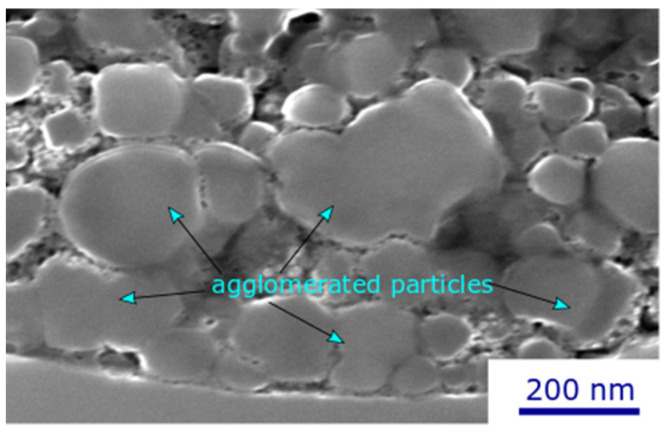
High-magnification SEM micrograph of agglomerated particles in a TiO_2_ deposit obtained with 70 wt% (condition C4).

**Figure 8 materials-17-01493-f008:**
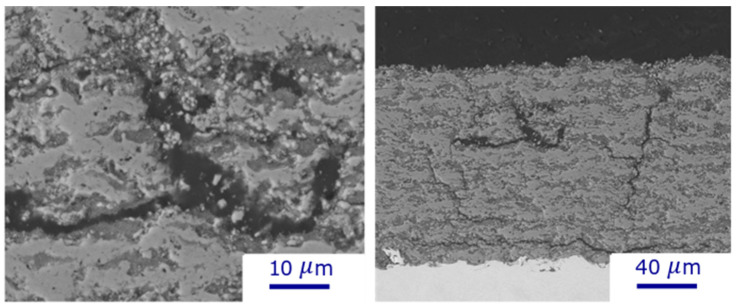
SEM micrographs of TiO_2_ deposits obtained with high solid concentrations (>55 wt%), high torch power (>70 kW), and low rotating sample holder speed (60 RPM) (condition C5).

**Figure 9 materials-17-01493-f009:**
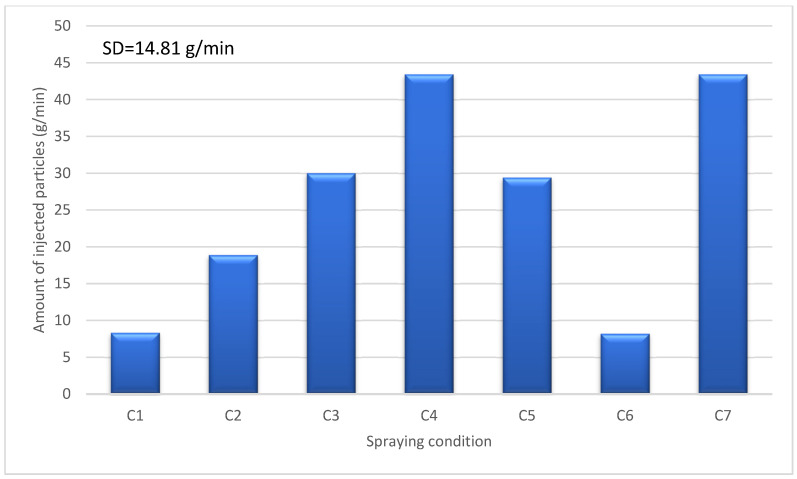
Number of injected particles (g/min) for different spraying conditions.

**Figure 10 materials-17-01493-f010:**
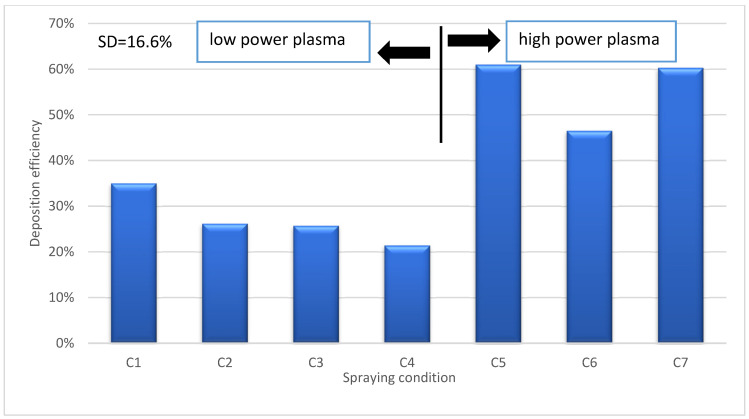
Deposition efficiency for different spraying conditions.

**Figure 11 materials-17-01493-f011:**
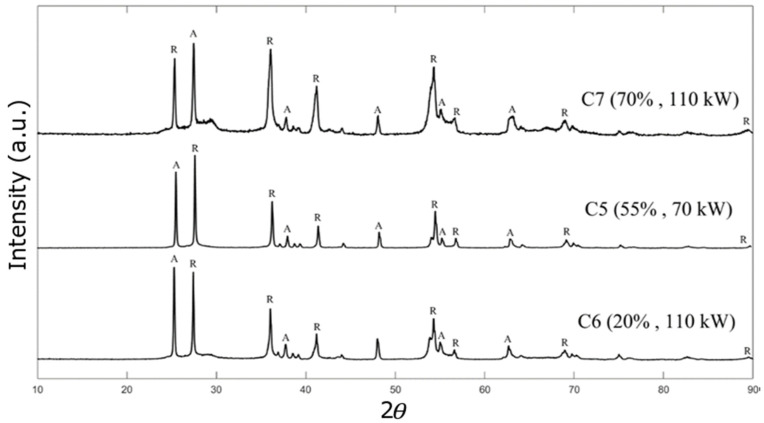
XRD patterns of the TiO_2_ deposits for conditions C5 to C7 (plasma power of 70 and 110 kW). A represents anatase. R represents rutile.

**Table 1 materials-17-01493-t001:** Spray conditions for the 3 MB torch.

Sample	Plasma Power (kW)	Gas Flow Rate (SLPM)	Current (A)	Suspension Concentration (wt%)	Feed Rate (mL/min)	Density (g/cm^3^)	Spray Time (s)	Rotating Sample Holder Speed (RPM)	Coating Thickness (μm)
C1	33	Ar/H_2_	600	20	34–36	1.19	45	60	55
C2				40	32–34	1.43	30		115
C3	(50, 3.5)			55	30–32	1.76	15		100
C4				70	28–30	2.14	10		70

**Table 2 materials-17-01493-t002:** Spray conditions for the Axial III torch.

Sample	Plasma Power (kW)	Gas Flow Rate (SLPM)	Total Gas Flow Rate (SLPM)	Suspension Concentration (wt%)	Flow Rate (mL/min)	Density (g/cm^3^)	Spray Time (s)	Rotating Sample Holder Speed (RPM)	Coating Thickness (μm)
C5	70	Ar/N_2_/H_2_	180	55	30–33	1.7	10	60	140
	45/45/10								
C6	110	Ar/N_2_/H_2_	220	20	34–36	1.17	30	180	55
C7	45/45/10			70	28–30	2.14	10		120

**Table 3 materials-17-01493-t003:** Percentage of anatase and rutile phases for the deposits produced with conditions C1 to C7.

Samples	Anatase (%)	Rutile (%)
C1	59.4	40.6
C2	52.8	47.2
C3	50.3	49.7
C4	66.9	33.1
C5	35.7	64.2
C6	48.1	51.9
C7	38.2	63.9

## Data Availability

Data are contained within the article.

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
