# Peer review of "Microstructure of Deposits Sprayed by a High Power Torch with Flash Boiling Atomization of High-Concentration Suspensions"

_materials, 2024, doi:10.3390/ma17071493_

Round 1

Reviewer 1 Report

Comments and Suggestions for Authors

The novelty of this work is the adoption of flash boiling atomisation for the preparation of nano-sized particles and the authors have demonstrated its application to titania synthesis.

The adoption of flash boiling atomisation was expected to avoid both plasma cooling and plasma disturbance, however very direct evidence for the absence of plasma cooling and plasma disturbance during the process was not demonstrated. The authors need to clarify and highlight the supporting data for the alleviation of plasma cooling and plasma disturbance. 

In addition to the control of process parameters such as concentration, flow rate and spray time, physicochemical mechanism discussions on the quenching of plasma cooling and plasma disturbance need to be addressed.

 Minor correction : 1) in table 1 and 2, g/cm3 can be corrected g/cm3

In table 1, for samples C1-C4, gas flow rate summation does not make 100 %.

Author Response

Hello,

Thank you for your constructive comments and feedback. Please see the attachment.

Reviewer 2 Report

Comments and Suggestions for Authors

Thank you for involving me for the reviewing of the manuscript entitled "Microstructure of Deposits Sprayed by a High Power Torch with Flash Boiling Atomization of High-concentration Suspensions". This work used flash boiling atomization as a new method to inject suspensions with high solid content into the high-power plasma flow to investigate the effect of torch power on deposit properties such as deposit microstructure, deposition efficiency and phase composition. In general, it is a nice work, but there are some detailed suggestions as follows:    
1) Reduce the number of keywords, it's too much.
2)There are many blank spaces in page 5, 8, 9 and 12, it is recommended to rearrange the layout.
3) Figures 6 and 8 should provide an SEM image with a higher magnification, e.g. 5 microns.
4) The conclusion suggests adopting a structured form, such as:    
1.……………….     
2. ………….

Author Response

(The authors gave the same response as above.)

Reviewer 3 Report

Comments and Suggestions for Authors

The flash boiling atomization is a new application in the SPS coating to improve the efficiency of solid content of the coating layer.  The objective is clear and the description of the result and discussion is well organized. Reviewer suggests to revise the manuscript considering the following comments;

- Describe the TiO2 coating in the introduction in detail. For example, describe the advantage of TiO2 and application of TiO2 coating.

- Indicate the non-melted particles and fully melted regions using arrow or dotted area in Fig. 5 and 6

- You'd better mention that "strain-tolerance" can be improved by the effect of vertical cracks at high solid content. 

Author Response

(The authors gave the same response as above.)

Reviewer 4 Report

Comments and Suggestions for Authors

The paper is interesting and useful, well-structured, and well-readable. The authors described the microstructure of TiO2 deposited from suspension by plasma spraying. Despite the idea of such a study is not entirely new, the manuscript does contain novel results. In my opinion, the authors in an interesting way presented the study on data for SPS TiO2 coatings. The obtained results are promising. The analysis and area of investigation are sufficient and well-discussed. The presented data are reliable and useful. The topics presented in this manuscript fully match MATERIALS. The title of the manuscript is satisfactory. Minor comments do not reduce the value of the article.

l. 136 - delamination - Please explain why it occurs. Short spray times limit the application.

l. 148 - Mettech torch – better Axial III gun (such a description is clear)

l. 150 - Mettech torch – as above.

l. 150 – Please add at the end of the paragraph;

1. Whether the temperature of the samples was measured during spraying?

2. How many times each experiment was repeated?

l. 152 – Table 1 – Please add the thickness of the coatings.

l. 159 - Mettech torch – as earlier.

l. 159 – Table 2 – Please add the thickness of coatings.

l. 229 – 3.1. Microstructure of TiO2………. - Figures and discussion of the surface microstructure would be a valuable addition.

l. 246 - Mettech 110 kW torch – as earlier.

l. 275 - Figure 5. – Please add at the end of the description “(condition 6)”.

l. 351 – Figure 9 illustrates – Please add standard deviations.

l. 360 - Figure 10 shows – Please add standard deviations.

l. 400 – Mettech gun at 70 & 110 kW– as earlier.

Author Response

(The authors gave the same response as above.)

Reviewer 5 Report

Comments and Suggestions for Authors

The authors gave a comprehensive introduction for the spray deposition. They introduced a new method, called flash boiling atomization, and explained clearly the working principle. They differentiated this new method with the conventional method for the plasma spraying deposition. After carefully reading, few comments and suggestions are raised.

1.      Please explain why the authors have to use two different torches. What are the major differences between both?

2.      Their spray times were short. Who did they know the suspension mixture was homogeneous when the suspension left the nozzle?

3.      Please explain the logic behind how the authors designed their experimental spray conditions.

4.      Why did the authors add nitrogen gas into the high plasma power condition, but did not consist of nitrogen gas into the lower plasma power condition? What is the major function or purpose for the nitrogen input?

5.      There is a labelling error in C7 of the Fig 11.

6.      Why did you add the capital G in the end of references 4, 5 and 6.

7.      Reference 18 is your new publication and does not have volume and page number. In addition to your original description, I will suggest using the doi (Digital Object Identifier).

Author Response

(The authors gave the same response as above.)

Round 2

Reviewer 1 Report

Comments and Suggestions for Authors

The revised version reflects the comments of the reviewers.